# ON ALIGNMENT IN DEEP LINEAR NEURAL NETWORKS

## ABSTRACT

We study the properties of *alignment*, a form of implicit regularization, in linear neural networks under gradient descent. We define alignment for fully connected networks with multidimensional outputs and show that it is a natural extension of alignment in networks with 1-dimensional outputs as defined by Ji and Telgarsky, 2018. While in fully connected networks, there always exists a global minimum corresponding to an aligned solution, we analyze alignment as it relates to the training process. Namely, we characterize when alignment is an *invariant* of training under gradient descent by providing necessary and sufficient conditions for this invariant to hold. In such settings, the dynamics of gradient descent simplify, thereby allowing us to provide an explicit learning rate under which the network converges linearly to a global minimum. We then analyze networks with layer constraints such as convolutional networks. In this setting, we prove that gradient descent is equivalent to projected gradient descent, and that alignment is impossible with sufficiently large datasets.

## 1 INTRODUCTION

Although overparameterized deep networks can interpolate randomly labeled training data (Du et al., 2019; Wu et al., 2019), training overparameterized networks with modern optimizers often leads to solutions that generalize well. This suggests that there is a form of *implicit regularization* occurring through training (Zhang et al., 2017).

As an example of implicit regularization, the authors in Ji & Telgarsky (2018) proved that the layers of linear neural networks used for binary classification on linearly separable datasets become aligned in the limit of training. That is, for a linear network parameterized by the matrix product $W_d W_{d-1} \ldots W_1$, the top left/right singular vectors $u_i$ and $v_i$ of layer $W_i$ satisfy $|v_{i+1}^T u_i| \to 1$ as the number of gradient descent steps goes to infinity.

Alignment of singular vector spaces between adjacent layers allows for the network representation to be drastically simplified (see Equation 3); namely, the product of all layers becomes a product of diagonal matrices with the exception of the outermost unitary matrices. If alignment is an invariant of training, then optimization over the set of weight matrices reduces to optimization over the set of singular values of weight matrices. Thus, importantly, alignment of singular vector spaces allows for the gradient descent update rule to be simplified significantly, which was used in Ji & Telgarsky (2018) to show convergence to a max-margin solution.

In this work, we generalize the definition of *alignment* to the multidimensional setting. We study when alignment can occur and moreover, under which conditions it is an invariant of training in linear neural networks under gradient descent. Prior works (Gidel et al., 2019; Saxe et al., 2014; 2019) have implicitly relied on invariance of alignment as an assumption on initialization to simplify training dynamics for 2 layer networks. In this work, we provide necessary and sufficient conditions for when alignment is an invariant for networks of arbitrary depth. Our main contributions are as follows:

1. We extend the definition of alignment from the 1-dimensional classification setting to the multi-dimensional setting (Definition 2) and characterize when alignment is an invariant of training in linear fully connected networks with multi-dimensional outputs (Theorem 1).
2. We demonstrate that alignment is an invariant for fully connected networks with multidimensional outputs only in special problem classes including autoencoding, matrix factorization

and matrix sensing. This is in contrast to networks with 1-dimensional outputs, where there exists an initialization such that adjacent layers remain aligned throughout training under *any* real-valued loss function and any training dataset.

3. Alignment largely simplifies the analysis of training linear networks: We provide an explicit learning rate under which gradient descent converges linearly to a global minimum under alignment in the squared loss setting (Proposition 1).

4. We prove that alignment cannot occur, let alone be invariant, in networks with constrained layer structure (such as convolutional networks), when the amount of training data dominates the dimension of the layer structure (Theorem 3).

5. We support our theoretical findings via experiments in Section 6.

As a consequence, our characterization of the invariance properties of alignment provides settings under which the gradient descent dynamics can be simplified and the implicit regularization properties can be fully understood, yet also shows that further results are required to explain implicit regularization in linear neural networks more generally.

## 2  RELATED WORK

Implicit regularization in overparameterized networks has become a subject of significant interest (Gunasekar et al., 2018a;b; Martin & Mahoney, 2018; Neyshabur et al., 2014). In order to characterize the specific form of implicit regularization, several works have focused on analyzing deep *linear* networks (Arora et al., 2019b; Gunasekar et al., 2018b; 2017; Soudry et al., 2018). Even though such networks can only express linear maps, parameter optimization in linear networks is non-convex and is studied in order to obtain intuition about optimization of deep networks more generally.

One such form of implicit regularization is *alignment*, identified by Ji & Telgarsky (2018) to analyze linear fully connected networks with 1-dimensional outputs trained on linearly separable data. They proved that in the limit of training, each layer, after normalization, approaches a rank 1 matrix, i.e.

$$\lim_{t \to \infty} \frac{W_i^{(t)}}{\|W_i^{(t)}\|_F} = u_i v_i^T$$

and that adjacent layers, $W_{i+1}$ and $W_i$ become *aligned*, i.e. $|v_{i+1}^T u_i| \to 1$.

In addition, Ji & Telgarsky (2018) proved that alignment in this setting occurs concurrently with convergence to the max-margin solution. Follow-up work mainly focused on this convergence phenomenon and gave explicit convergence rates for overparameterized networks trained with gradient descent (Arora et al., 2019c; Zou et al., 2018).

Our definition of invariance of alignment extends assumptions on initialization appearing in various prior works (Gidel et al., 2019; Saxe et al., 2014; 2019). While the connection to alignment was not mentioned in their work, the authors in Gidel et al. (2019) begin to generalize alignment to multidimensional outputs by considering two-layer networks initialized so that layers are aligned with each other and to the data. We generalize this to networks of any depth, showing that our definition of alignment corresponds to the initialization considered in Gidel et al. (2019). Moreover, we establish necessary and sufficient conditions for when alignment is an invariant of training in Theorem 1 instead of assuming these conditions. Furthermore, their result on sequential learning of components can be derived via our singular value update rule in Corollary 1.

Balancedness is another closely related form of implicit regularization in linear neural networks. It was introduced in Arora et al. (2018) and defined as the property that if $W_i^T W_i = W_{i+1} W_{i+1}^T$ for all $i$ at initialization, then this property is invariant under gradient flow. Du et al. (2018) present a more general form, that $W_i^T W_i - W_{i+1} W_{i+1}^T$ is constant under gradient flow. In practice, analyses rely on this quantity being close to or exactly zero. In this exact setting, balancedness indeed implies alignment of singular vector spaces between consecutive layers. To study gradient descent, slightly more general notions such as approximate balancedness (Arora et al., 2019a) and $\epsilon$-balancedness have been introduced. Du et al. (2018) also defined balancedness with respect to convolutional networks, showing that under gradient flow, the difference in the norm of the weights of consecutive layers is an invariant. Generally, the goal of identifying invariants of training such as balancedness or alignment is to help understand both the dynamics of training and properties of solutions at the end of training.

## 3    Definition of Alignment in the Multi-dimensional Setting

In this section, we first define alignment for linear neural networks with multi-dimensional outputs. We then define when alignment is an invariant of training.

We consider *linear* neural networks. Let $f : \mathbb{R}^{k_0} \to \mathbb{R}^{k_d}$ denote such a $d$-layer network, i.e.

$$f(x) = W_d W_{d-1} \ldots W_1 x, \tag{1}$$

where $W_i \in \mathbb{R}^{k_i \times k_{i-1}}$ for $i \in [d]$, where we follow the convention that $[d] = \{1, 2, \ldots d\}$. Let $(X, Y) \in \mathbb{R}^{k_0 \times n} \times \mathbb{R}^{k_d \times n}$ denote the set of training data pairs $\{(x^{(i)}, y^{(i)})\}$ for $i \in [n]$. Gradient descent with learning rate $\gamma$ is used to find a solution to the following optimization problem:

$$\arg\min_{f \in \mathcal{F}} \frac{1}{2n} \sum_{i=1}^{n} \ell(f(x^{(i)}), y^{(i)}), \tag{2}$$

where $\mathcal{F}$ is the set of linear functions represented by $f$ and $\ell$ is a real-valued loss function. When not stated otherwise, we assume $\ell(f(x^{(i)}), y^{(i)}) = \|y^{(i)} - f(x^{(i)})\|_2^2$, which is the squared loss (MSE). In addition, we denote by $W_i^{(t)}$ for $t \in \mathbb{Z}_{\geq 0}$ the weight matrix $W_i$ after $t$ steps of gradient descent. When there are no additional constraints on the matrices $W_i$, then $f$ is a fully connected network.

We next introduce a generalized form of the singular value decomposition:

**Definition 1.** *An **unsorted, signed singular value decomposition (usSVD)** of a matrix $A \in \mathbb{R}^{m \times n}$ is a triple $U \in \mathbb{R}^{m \times m}, \Sigma \in \mathbb{R}^{m \times n}, V \in \mathbb{R}^{n \times n}$ such that $U, V$ are orthonormal matrices, $\Sigma$ is diagonal, and $A = U\Sigma V^T$.*

In contrast to the usual definition of singular value decomposition (SVD) of a matrix, the diagonal entries of $\Sigma$ may be in any order and take negative values. Throughout, we will refer to the entries of $\Sigma$ in a usSVD as singular values and the vectors in $U, V$ as singular vectors. Using the usSVD, we now generalize the notion of alignment from Ji & Telgarsky (2018) to the multi-dimensional setting.

**Definition 2.** *Let $f = W_d W_{d-1} \ldots W_1$ be a linear network. We say that $f$ is **aligned** if there exists a usSVD $W_i = U_i \Sigma_i V_i^T$ with $U_i = V_{i+1}$ for all $i \in [d-1]$. (We also say that a matrix $A$ is aligned with another matrix $B$ if there exist usSVD's $A = U_A \Sigma_A V_A^T, B = U_B \Sigma_B V_B^T$ such that $V_A = U_B$.)*

Note that if $W_i$ and $W_{i+1}$ are rank 1 matrices in an aligned network $f$, then the inner product of the first columns of $V_{i+1}$ and $U_i$ is 1 in absolute value. Hence Definition 2 is consistent with alignment in the 1-dimensional setting from Ji & Telgarsky (2018).

We next define when alignment is an invariant of training for deep linear networks. Again, such invariants are of interest since they may provide insights into properties of trained networks and significantly simplify the dynamics of gradient descent.

**Definition 3.** *Alignment is an invariant of training for a linear neural network $f$ if there exists an initialization $\{W_j^{(0)}\}_{j=1}^d$ such that $W_1^{(\infty)}, W_2^{(\infty)}, \ldots, W_d^{(\infty)}$ achieves zero training error [1] and for all gradient descent steps $t \in \mathbb{Z}_{\geq 0}$*

*(a) the network $f$ is aligned;*

*(b) $W_i^{(t)} = U_i \Sigma_i^{(t)} V_i^T$ for all $i \in \{2, \ldots d-1\}$, that is, $U_i, V_i$ are not updated;*

*(c) $W_1^{(t)} = U_1 \Sigma_1^{(t)} V_1^{(t)^T}$ and $W_d^{(t)} = U_d^{(t)} \Sigma_d^{(t)} V_d^T$, that is, $U_1$ and $V_d$ are not updated.*

*If additionally, $V_1$ and $U_d$ are not updated for any $t \in \mathbb{Z}_{\geq 0}$, then we say that **strong alignment is an invariant of training**.*

When alignment is an invariant of training, there are important consequences for training. In particular, note that when the network $f$ is aligned with usSVDs $W_i = U_i \Sigma_i V_i^T$ for all $1 \leq i \leq d$, then

$$f(x) = W_d \cdots W_1 x = U_d \left( \prod_{i=0}^{d-1} \Sigma_{d-i} \right) V_1^T x. \tag{3}$$

---

[1]The interpolation condition in this definition (i.e., achieving zero training error) is important in ruling out several architectures where the layers are trivially aligned. For example, if all layers are constrained to be diagonal matrices throughout training, then the layers are all trivially aligned, but cannot interpolate datasets where the target is not the product of a diagonal matrix with the input.

Hence if alignment is an invariant of training, then the singular vectors of layers 2 through $d - 1$ are never updated and the analysis of gradient descent can be limited to the singular values of the layers and the matrices $V_1$ and $U_d$.

**Remarks.** For the remainder of the paper, we assume that the gradient of the loss function at initialization $\{W_i^{(0)}\}_{i=1}^d$ is non-zero. Otherwise, training with gradient descent would not proceed. We also only consider datasets $(X, Y)$ for which there is a linear network that achieves loss zero. This is consistent with the assumptions in Ji & Telgarsky (2018).

## 4 ALIGNMENT IN FULLY CONNECTED NETWORKS

In this section, we first characterize when alignment is an invariant of training for fully connected networks (Theorem 1). In particular, we show that this is not the case in general. We then present special classes of problems for which alignment is an invariant of training, namely autoencoding, matrix factorization, and matrix sensing. In contrast, for a linear network with 1-dimensional outputs, we demonstrate that there exists an initialization for which the layers remain aligned throughout training given any dataset and any real-valued loss function. Finally, we discuss various consequences of alignment, including a proof of linear convergence of gradient descent to an interpolating solution.

### 4.1 CHARACTERIZATION OF ALIGNMENT WITH MULTI-DIMENSIONAL OUTPUTS

Theorem 1 is one of our main results and characterizes when alignment is an invariant of training in a fully connected network with multi-dimensional outputs. To simplify notation, we consider the case when the layers are square matrices, i.e. $k_i = k_j$ for all $0 \leq i, j \leq d$. The general result for non-square matrices is provided in Appendix D.

**Theorem 1.** *Let $f : \mathbb{R}^k \to \mathbb{R}^k$ be a linear fully connected network with $d \geq 3$ square layers of size $k > 1$. Alignment is an invariant of training under the squared loss on a dataset $(X, Y) \in \mathbb{R}^{k \times n} \times \mathbb{R}^{k \times n}$ if and only if there exist orthonormal matrices $U, V \in \mathbb{R}^{k \times k}$ such that $U^T Y X^T V$ and $V^T X X^T V$ are diagonal.*

The full proof of this result is presented in Appendices A-E; here, we provide a proof sketch.

*Proof Sketch.* The proof essentially follows by induction. For the base case, we initialize the layers $\{W_i\}_{i=1}^d$ to satisfy the conditions for alignment given in Definition 3. Assuming that these conditions hold at gradient descent step $t$, we prove that they hold at step $t + 1$.

After substituting the alignment conditions into the gradient descent update equation for the squared loss at step $t+1$ and cancelling terms, we obtain that alignment is an invariant of training if and only if

$$U_d^{(t)T} \sum_{k=1}^n (y^{(k)} - f(x^{(k)})) x^{(k)T} V_1^{(t)} \tag{4}$$

is a diagonal matrix. By considering the update for $W_1^{(t)}$ and $W_d^{(t)}$, one sees that alignment implies strong alignment and so $U_d, V_1$ are also invariant across updates. Thus, let $U_d = U$ and $V_1 = V$. By expanding $f(x^{(k)})$ using equation 3, and considering the update across multiple timesteps, we obtain that the matrix in equation 4 is diagonal if and only if $U^T Y X^T V$ and $V^T X X^T V$ are diagonal. To complete the proof, we show in Appendix D that under strong alignment, gradient descent converges to a solution with zero training error. $\square$

Theorem 1 implies that invariance of alignment throughout training holds only for special classes of problems. In particular, the above implies that alignment is an invariant of training when $X$ and $Y$ have the same right singular vectors, a very special condition on the data. Note that this corresponds to the $\epsilon = 0$ data condition with the initialization considered in Gidel et al. (2019). In Section 6, we also provide empirical support showing that alignment is not an invariant of training for important tasks that violate the data condition presented here, such as multi-class classification.

## 4.2 CLASSES OF PROBLEMS WITH ALIGNMENT

We next discuss classes of problems for which alignment is an invariant of training.

**Autoencoding:** In the case when $X = Y$, it holds that $U^T Y X^T V = U^T X X^T V$. Taking $U = V$ to be the left singular vectors of $X$ satisfies the conditions of Theorem 1.

**Matrix Factorization and Inversion**: In the case of matrix factorization, we have that $X = I$. Hence taking $U$ and $V$ to be the left and right singular vectors of $Y$ respectively satisfies the conditions of Theorem 1. For matrix inversion, we have that $Y = I$ and we proceed analogously.

**Matrix Sensing.** Given pairs of observations $\{(M_i, y_i)\}_{i=1}^n$ with $M_i \in \mathbb{R}^{k \times k}$ and $y_i = \text{Tr}(M_i^T X^*)$ for some unobserved matrix $X^* \in \mathbb{R}^{k \times k}$, gradient descent on $\{W_i\}_{i=1}^n$ is used to solve

$$\arg \min_{\{W_i\}} \frac{1}{2n} \sum_{i=1}^n \|y_i - \text{Tr}(M_i^T W_d W_{d-1} \ldots W_1)\|_2^2.$$

Implicit regularization in the matrix sensing setting has been analyzed extensively (Arora et al., 2019b; Du et al., 2018; Gunasekar et al., 2017; Li et al., 2018). Theorem 1 shows that alignment is an invariant of training in this setting if and only if $M_i = U \Lambda_i V^T$ for all $i \in [n]$, and $U_d = U, V_1 = V$.

**1-dimensional Outputs.** In Appendix F, we show that alignment is an invariant of training for fully connected networks with 1-dimensional outputs for any real-valued loss function provided that gradient descent converges to zero training error.

## 4.3 CONSEQUENCES OF ALIGNMENT

We next discuss various consequences of the invariance of alignment for the analysis of training. Our explicit characterization of alignment as an invariant is significant as it allows us to greatly simplify the convergence analysis of gradient descent, which is a main goal of defining an invariant of training.

The following corollary (proof in Appendix B) follows from the proof of Theorem 1, and shows that under alignment the gradient descent update rule is simplified significantly.

**Corollary 1.** *Let $r = \min(k_0, k_1, \ldots, k_d) > 1$ and let the top left $r \times r$ submatrix of $U^T Y X^T V$ be $\Lambda'$ and that of $V^T X X^T V$ be $\Lambda$. Under the invariance of strong alignment (i.e., when $\Lambda'$ and $\Lambda$ are diagonal), we can express the partial derivative with respect to $W_i$ as follows:*

$$\frac{\partial L}{\partial W_i} = -\frac{1}{n} U_i \left( \prod_{j=i+1}^d \Sigma_j{}^T (U^T Y X^T V_1 - \Sigma_d \cdots \Sigma_1 V^T X X^T V) \prod_{j=1}^{i-1} \Sigma_j{}^T \right) V_i^T. \tag{5}$$

*As a result, gradient descent only updates the first $r$ values of $\Sigma_i$. Let $\Sigma'_i{}^{(t)}$ be the top left $r \times r$ matrix of $\Sigma_i^{(t)}$. The updates are then given by:*

$$\Sigma'_i{}^{(t+1)} = \Sigma'_i{}^{(t)} + \frac{\gamma}{n} \prod_{j=1}^d \Sigma'_j{}^{(t)} (\Lambda' - \prod_{j=1}^d \Sigma'_j{}^{(t)} \Lambda). \tag{6}$$

*The other entries of $\Sigma_i^{(t+1)}$ are not updated.*

We can use this corollary to provide an explicit learning rate under which gradient descent converges linearly to a global minimum. The proof of the following proposition is given in Appendix C.

**Proposition 1.** *For $k \in [r]$, let $\sigma_k(W_i)$ denote the $k$th entry of $\Sigma_i$ in the usSVD of $W_i$, and let $\lambda_k$, $\lambda'_k$ denote the $k$th entries of $\Lambda$, $\Lambda'$ respectively. Under the conditions of Corollary 1 and assuming that $\sigma_k(W_i^{(0)}) > 0$ and $\prod_{i=1}^d \sigma_k(W_i^{(0)}) < \frac{\lambda'_k}{\lambda_k}$ for all $k \in [r]$, if the learning rate satisfies $\gamma \leq \frac{n \ln 2}{d} \cdot \min_k \frac{\sigma_k(W_i^{(0)})^2 \lambda_k}{\lambda_k'^2}$ then gradient descent only updates the top $r$ singular values of the solution and converges linearly to the global minimum.*

**Remarks.** Shamir (2018) shows that for linear neural networks with one-dimensional outputs, the rate of convergence can be exponentially slow in the depth. Proposition 1 shows that for a fixed depth, gradient descent converges linearly. Our analysis in Appendix C shows that our upper bound on the rate of convergence can also grow exponentially in the depth.

**Alignment in the Limit of Training.** We briefly comment on understanding whether alignment will occur in the limit of training. The following proposition, which states that for a 2-layer network, an aligned solution achieves the minimum $\ell_2$-norm. The proof is given in Appendix G.

**Proposition 2.** *Let $W_1, W_2$ be matrices such that $W_2 W_1 = P$, for a fixed matrix $P$. Then, $\|W_1\|_F^2 + \|W_2\|_F^2$ achieves a minimum at the solution where $W_1$ and $W_2$ are aligned and 0-balanced, i.e. there exist usSVD's $W_1 = W\Sigma V^T, W_2 = U\Sigma W^T$.*

It has been shown that SGD in the overparameterized setting for a network initialized close to zero will converge to a solution close in $\ell_2$-norm to the minimum $\ell_2$-norm solution (Azizan et al., 2019). Therefore we expect such networks to converge to a solution which is close to an aligned solution.

# 5  ALIGNMENT UNDER GENERAL LAYER STRUCTURE

In the previous section, we analyzed fully connected networks, where parameters of each weight matrix are optimized independently. The most commonly used deep learning models, however, rely on convolutional layers or layers with other forms of constraints. In this section, we analyze alignment in the setting of linear networks with layer constraints. In particular, we show that when the dimension of the subspace induced by the layer constraints is small compared to the number of training samples, alignment cannot happen, let alone be an invariant of training.

## 5.1  LINEAR NEURAL NETWORKS WITH LAYER STRUCTURE

We start by setting up mathematical terminology to describe different layer structures.

**Definition 4.** *Let $S \subset \mathbb{R}^{m \times n}$ be a linear subspace of matrices and let $\{A_i\}_{i=1}^r$ be an orthogonal[2] basis for $S$. Layer $W_i$ has layer structure $S$ if $W_i \in S$, i.e., there exist coefficients $\{c_j^i\}_{j=1}^r \subset \mathbb{R}$ such that $W_i = \sum_{j=1}^r c_j^i A_j$, and gradient descent operates on the $\{c_j^i\}_{i,j=1}^r$.*

Definition 4 encompasses layer structures commonly used in practice, such as:

*Convolutional layers:* Treating a $p \times p$ image as a vector in $\mathbb{R}^{p^2}$, a single $s \times s$ convolutional filter with stride 1 and padding $(s-1)/2$ maps the image to another $p \times p$ image; this linear transformation is a matrix in $\mathbb{R}^{p^2 \times p^2}$ and the set of all such transformations forms an $s^2$-dimensional subspace. The parameters of the filter are coefficients of an orthogonal basis of this subspace; see Appendix I.

*Layers with Sparse Connections:* Consider a fixed connection pattern between layers such that the $j^{th}$ hidden unit in layer $i$ depends only on a subset of units in layer $i-1$. In this case, the subspace $S$ consists of matrices where particular entries are forced to be zero corresponding to missing connections between features in consecutive layers.

The following theorem provides, in closed-form, the gradient descent update rules for linear networks with layer structure. The proof is provided in Appendix H.

**Theorem 2.** *Performing gradient descent on the basis coefficients $\{c_j^i\}_{j=1}^r$ leads to the following weight matrix updates:*

$$W_i^{(t+1)} = W_i^{(t)} - \eta \cdot \pi_S \left( \frac{\partial l}{\partial W_i^{(t)}} \right),$$

*where $\pi_S$ denotes the projection operator onto $S$.*

Theorem 2 shows that gradient descent in networks with layer structure is equivalent to projected gradient descent[3]. Hence alignment is an invariant of training if and only if it holds throughout the projected gradient descent updates and leads to an aligned solution with zero training loss.

---

[2]Orthogonality is w.r.t the inner product $\langle A, B \rangle = \text{Tr}(A^T B)$, or equivalently the dot product in $\mathbb{R}^{mn}$

[3]$\pi_S$ is a projection in the traditional sense if and only if the $A_j$ form an orthonormal basis; otherwise, $\pi_S$ is a projection onto $S$ followed by an appropriate scaling in each basis direction.

## 5.2 Necessary Condition for Alignment

Motivated by the above characterization via projected gradient descent, we now show that for layer structures with constrained dimension, aligned networks generally cannot achieve zero training error under the squared loss, given sufficient data (Proposition 4). This is the case even when there is a solution with the desired layer structure that achieves zero training error. Hence, if loss is minimized to zero, gradient descent must lead to a non-aligned network.

We first show that for an aligned network which interpolates the data, the first and last layer must align with the pseudoinverse. The proof of this result is presented in Appendix J.

**Proposition 3.** *Let $(X, Y) \in \mathbb{R}^{k_0 \times n} \times \mathbb{R}^{k_d \times n}$ such that $n \geq k_0$ and $X$ is full-rank (ensuring that $XX^T$ is invertible). If an aligned network $f = W_d W_{d-1} \ldots W_1$ achieves zero error under squared loss (i.e. if $Y = f(X)$), then $W_d^T$ aligns with $YX^T(XX^T)^{-1}$, which in turn aligns with $W_1^T$.*

The following result tells us that when a linear space $\mathcal{S}$ of matrices is sufficiently low-dimensional, the set of matrices that align with an element of $\mathcal{S}$ has measure zero. While we are mainly interested in the setting where $n \geq k$, we state it in full generality using $\binom{m}{2} = 0$, when $m < 2$.

**Proposition 4.** *Let $\mathcal{S}$ be an $r$-dimensional linear subspace of $k \times k$ matrices. If $r < k - 1 - \binom{k-n}{2}$ then the set of matrices of size $k \times n$ that can align with an element of $\mathcal{S}$, excluding scalar multiples of the identity, has Lebesgue measure zero.*

The proof of Proposition 4 is provided in Appendix K. Taken together, Propositions 3 and 4 imply Theorem 3, which states that alignment does not occur in linear networks with constrained layer structures given enough training samples. We assume $k = k_0 = \cdots = k_d$ and that all layers have the same structure, $\mathcal{S}$. The statement can trivially be extended to the general setting.

**Theorem 3.** *Let $n \geq k$, let $X, Y \in \mathbb{R}^{k \times n}$ be generic, let $\mathcal{S} \subset \mathbb{R}^{k \times k}$ be a linear subspace of dimension $r < k - 1$, and let $W_1, \ldots, W_d \in \mathcal{S}$ such that at least one $W_i$ is not a scalar multiple of the identity[4]. If the network $f = W_d \cdots W_1$ satisfies $Y = f(X)$, then $f$ is not aligned.*

Theorem 3 is in contrast to fully connected networks (i.e., no layer constraints), where we showed that alignment is possible for particular classes of problems including autoencoders. An explicit example of a convolutional linear autoencoder, where alignment is ruled out by Theorem 3, is discussed next.

**Example.** *If $m \geq 4$, then a generic dataset consisting of $n \geq m^2$ $m \times m$ images cannot be aligned by any convolutional linear autoencoder with filter size 3, aside from the trivial case where all layers are scalar multiples of the identity. This follows from letting $k = m^2$, $r = 9$ in Proposition 4.*

## 6 Empirical Support

In this section, we provide experimental results to validate our findings[5]. We measure two properties: (1) invariance of alignment from initialization, and (2) alignment between layers. Invariance of alignment at time $t$ is measured by the average dot product between corresponding columns of $U_i^{(t)}$ and $U_i^{(0)}$, as well as $V_i^{(t)}$ and $V_i^{(0)}$. Alignment is measured by the average dot product between corresponding columns of $U_i^{(t)}$ and $V_{i+1}^{(t)}$. For both, a value of 1 is perfect alignment / invariance.

We begin by demonstrating that alignment is not an invariant of training for fully connected networks when the data conditions of Theorem 1 are violated. Figure 1a shows an example where alignment is not an invariant for multi-dimensional regression with random data under squared loss. We used standard normal inputs $X \in \mathbb{R}^{9 \times 9}$ and targets $Y \in \mathbb{R}^{9 \times 9}$, and a 2-hidden layer network initialized so that alignment holds at the start of training. Since $X$ and $Y$ do not have the same right singular vectors, the conditions of Theorem 1 are violated, and hence alignment is not an invariant of training. In Figures 1b and c, we show that alignment is also not an invariant in standard classification settings. We trained a 2-hidden layer fully connected network to classify a linearly separable subset of 256 MNIST examples under MSE loss and cross entropy loss. Figure 1b is consistent with the generalization of

---

[4]This is not a serious restriction; modulo scalar multiplication, the only case in which such a network could achieve zero loss is autoencoding, in which case the latent space would be a scalar multiple of the data itself.

[5]Hyperparameter settings are detailed in Appendix M

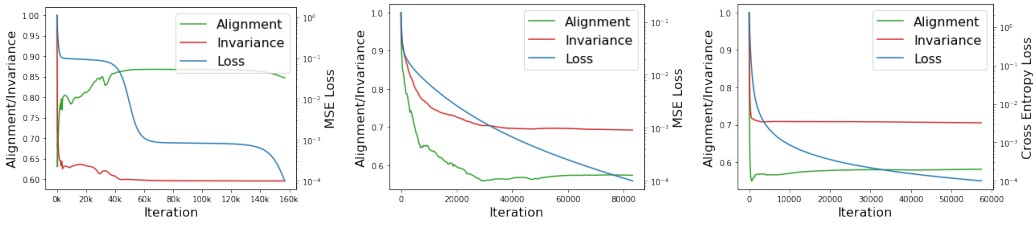

(a) Multi-dimensional regression on random data with squared loss.

(b) Multi-class classification on MNIST with squared loss.

(c) Multi-class classification on MNIST with cross entropy loss.

Figure 1: Examples of fully connected networks with multi-dimensional outputs where alignment is not an invariant of training.

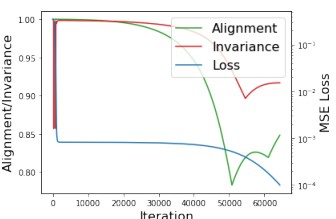

(a) Matrix factorization with layers constrained to be Toeplitz matrices.

(b) Autoencoding a single MNIST example using a convolutional network.

Figure 2: Examples of layer constrained networks, where alignment is not an invariant of training.

Theorem 1 (Appendix D). Interestingly, this result empirically transfers to the case of cross entropy loss, suggesting that our theoretical results may also be relevant for other loss functions.

In networks with constrained layer structure, Theorem 3 shows that given sufficient data alignment cannot occur. We now present empirical evidence that alignment is not an invariant of training, even when the number of training samples is much smaller than the output dimension of the network or the dimensionality of the layer structure is much larger than that of the output. In the setting of matrix factorization ($Y \in \mathbb{R}^{k \times k}$, $X = I$), $k = n$, so Theorem 3 states that alignment is impossible when the linear structure has dimension $r < k - 1$. In Figure 2a, we observe that alignment is not invariant also even when $r \geq k - 1$, by training a 2-hidden layer Toeplitz network to factorize a $4 \times 4$ matrix. Our network has 4 hidden units per layer and thus $r = 7, k = 4, n = 4$. Even when $n < r < k$, we observe that alignment is not an invariant. In Figure 2b, we show that alignment is not an invariant of training when autoencoding a single MNIST example using a 2-hidden layer linear convolutional network (i.e. $n = 1, r = 9, k = 784$). In Appendix L, we provide empirical validation that alignment is indeed an invariant of training when the data conditions of Theorem 1 are satisfied.

## 7 DISCUSSION

We generalized the definition of alignment to linear networks with multi-dimensional outputs. We then analyzed the invariance properties of alignment, showing that under particular data conditions alignment is an invariant for fully connected networks, which allows us to significantly simplify the convergence analysis of gradient descent. We then extended our analysis of alignment to networks with constrained layer structures, such as convolutions, and proved that alignment cannot be an invariant of training in such networks when the dimension of the layer structure $r$ is small compared to the number of training samples $n$.

While the simplification of gradient descent convergence analysis in the fully connected setting shows that our alignment definition is useful in understanding such networks, the fact that it does not generalize as an invariant to the constrained layer structure setting suggests that other approaches may be necessary to fully understand implicit regularization, such as studying how architecture influences the function classes that can be represented by deep networks (Savarese et al., 2019; Zhang et al., 2020; Radhakrishnan et al., 2019).

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

APPENDIX

A  OUTLINE OF PROOF FOR THEOREM 1, COROLLARY 1, AND PROPOSITION 1

We now provide an outline of our results and proofs.

1. In Appendix A, we introduce Lemmas 1, 2, which will be used to prove Theorem 1.

2. In Appendix B, we provide the proof of Corollary 1 - the simplification of gradient descent under alignment - which relies on Lemma 2.

3. In Appendix C, we provide the proof of Proposition 1 - linear convergence under strong alignment - which relies on Corollary 1.

4. In Appendix D, we introduce Theorem 4, which is a generalization of Theorem 1 to fully connected networks with rectangular layers. We use Lemma 2 and Proposition 1 to prove Theorem 4.

5. In Appendix E, we finally prove Theorem 1, which follows from Theorem 4.

Here, we present two lemmas that will be used extensively in our proofs.

Clearly strong alignment being an invariant implies that alignment is an invariant. Now we show that alignment implies strong alignment in the case of networks with square matrix layers.

**Lemma 1.** *Let $\{W_i\}_{i=1}^d \subset \mathbb{R}^{k \times k}$, where $d \geq 3$. If alignment is an invariant of training under the squared loss for network $f = W_d W_{d-1} \ldots W_1$ on data $(X, Y) \in \mathbb{R}^{k \times n} \times \mathbb{R}^{k \times n}$, then strong alignment is also invariant.*

*Proof.* Assume that alignment is an invariant of training. Gradient descent on the objective

$$\arg\min_{f \in \mathcal{F}} \frac{1}{2n} \sum_{i=1}^n \|y^{(i)} - f(x^{(i)})\|_2^2 \tag{7}$$

proceeds via the following update rule:

$$W_i^{(t+1)} = W_i^{(t)} + \frac{\gamma}{n}(W_d^{(t)} \ldots W_{i+1}^{(t)})^T \sum_{l=1}^n (y^{(l)} - f(x^{(l)}))(W_{i-1}^{(t)} \ldots W_1^{(t)} x^{(l)})^T, \quad \forall i \in [d]. \tag{8}$$

Since alignment is an invariant, the initialization satisfies $W_i^{(t)} = U_i \Sigma_i^{(t)} V_i^T$ for $2 \leq i \leq d-1$, $W_1^{(t)} = U_1 \Sigma_1^{(t)} V_1^{(t)^T}$, and $W_d^{(t)} = U_d^{(t)} \Sigma_d^{(t)} V_d^T$, where $U_i = V_{i+1}$ for $i \in [d-1]$. For $2 \leq i \leq d-1$, substituting into Equation (8) yields

$$W_i^{(t+1)} = U_i \Sigma_i^{(t)} V_i^T + \frac{\gamma}{n}(U_d^{(t)} \Sigma_d^{(t)} \cdots \Sigma_{i+1}^{(t)} V_{i+1}^T)^T \sum_{l=1}^n (y^{(l)} - f(x^{(l)}))(U_{i-1} \Sigma_{i-1}^{(t)} \cdots \Sigma_1^{(t)} V_1^{(t)^T} x^{(l)})^T$$

$$= U_i \left( \Sigma_i^{(t)} + \frac{\gamma}{n} \prod_{j=i+1}^d \Sigma_j^{(t)^T} U_d^{(t)^T} \sum_{l=1}^n (y^{(l)} - f(x^{(l)})) x^{(l)^T} V_1^{(t)} \prod_{j=1}^{i-1} \Sigma_j^{(t)^T} \right) V_i^T$$

$$= U_i \left( \Sigma_i^{(t)} + \frac{\gamma}{n} \prod_{j=i+1}^d \Sigma_j^{(t)^T} (U_d^{(t)^T} Y X^T V_1^{(t)} - \Sigma_d^{(t)} \cdots \Sigma_1^{(t)} V_1^{(t)^T} X X^T V_1^{(t)}) \prod_{j=1}^{i-1} \Sigma_j^{(t)^T} \right) V_i^T.$$

Since alignment is an invariant, the quantity

$$\prod_{j=i+1}^d \Sigma_j^{(t)^T} (U_d^{(t)^T} Y X^T V_1^{(t)} - \Sigma_d^{(t)} \cdots \Sigma_1^{(t)} V_1^{(t)^T} X X^T V_1^{(t)}) \prod_{j=1}^{i-1} \Sigma_j^{(t)^T} \tag{9}$$

is a diagonal matrix for all $t$. Since each of the $\Sigma_j$ are square, full rank matrices, the quantity

$$U_d^{(t)^T} Y X^T V_1^{(t)} - \Sigma_d^{(t)} \cdots \Sigma_1^{(t)} V_1^{(t)^T} X X^T V_1^{(t)}$$

must be diagonal for all $t$.

The update rule for $W_1$ is given by

$$W_1^{(t+1)} = W_1^{(t)} + \frac{\gamma}{n}(W_d^{(t)} \cdots W_2^{(t)})^T \sum_{l=1}^n (y^{(l)} - f(x^{(l)}))x^{(l)T}$$

$$U_1 \Sigma_1^{(t+1)} V_1^{(t+1)T} = U_1 \Sigma_1^{(t)} V_1^{(t)T} + V_2 \prod_{j=2}^d \Sigma_j^{(t)T} U_d^{(t)T}(YX^T - U_d \Sigma_d^{(t)} \cdots \Sigma_1^{(t)} V_1^{(t)T} XX^T)$$

$$\implies \Sigma_1^{(t+1)} V_1^{(t+1)T} V_1^{(t)} = \Sigma_1^{(t)} + \prod_{j=2}^d \Sigma_j^{(t)T}(U_d^{(t)T} YX^T V_1^{(t)} - \Sigma_d^{(t)} \cdots \Sigma_1^{(t)} V_1^{(t)T} XX^T V_1^{(t)}),$$

which is diagonal. Therefore $V_1^{(t+1)T} V_1^{(t)}$ is diagonal, and since this is also an orthogonal matrix we must have that $V_1^{(t+1)} = V_1^{(t)}$.

Similarly, the update rule for $W_d$ is given by:

$$W_d^{(t+1)} = W_d^{(t)} + \frac{\gamma}{n} \sum_{l=1}^n (y^{(l)} - f(x^{(l)}))x^{(l)T}(W_{d-1}^{(t)} \cdots W_1^{(t)})^T$$

$$U_d^{(t+1)} \Sigma_d^{(t+1)} V_d^T = U_d^{(t)} \Sigma_1^{(t)} V_d^T + (YX^T - U_d^{(t)} \Sigma_d^{(t)} \cdots \Sigma_1^{(t)} V_1^{(t)T} XX^T)V_1^{(t)} \prod_{j=1}^{d-1} \Sigma_j^{(t)T} U_{d-1}^{(t)T}$$

$$\implies U_d^{(t)T} U_d^{(t+1)} \Sigma_d^{(t+1)} = \Sigma_d^{(t)} + (U_d^{(t)T} YX^T V_1^{(t)} - \Sigma_d^{(t)} \cdots \Sigma_1^{(t)} V_1^{(t)T} XX^T V_1^{(t)}) \prod_{j=1}^{d-1} \Sigma_j^{(t)T} U_{d-1}^{(t)T},$$

which is diagonal. Therefore $U_d^{(t)T} U_d^{(t+1)}$ is also diagonal, implying that $U_d^{(t)} = U_d^{(t+1)}$. Therefore strong alignment is also an invariant. This means that alignment being an invariant and strong alignment being an invariant are equivalent in the setting where all the $k_i$ are equal. $\square$

Now that we have shown the equivalence of alignment being an invariant and strong alignment being an invariant in the setting where all the layers are square, we prove the following lemma for the general case where the $k_i$ are not necessarily all equal.

**Lemma 2.** *Let $f : \mathbb{R}^{k_0} \to \mathbb{R}^{k_d}$ be a linear fully connected network as in Equation equation 1, and let $r = \min(k_0, \ldots, k_n)$. For training under the squared loss on the dataset $(X, Y)$, there exists an aligned initialization $f(x) = W_d^{(0)} \cdots W_1^{(0)} x$ such that $W_i^{(t)} = U_i \Sigma_i^{(t)} V_i^T$ for all $i \in [d]$ (that is, $U_i, V_i$ are not updated) if and only if there exist orthonormal matrices $U \in \mathbb{R}^{k_d \times k_d}, V \in \mathbb{R}^{k_0 \times k_0}$ such that*

$$U^T YX^T V = \begin{bmatrix} \Lambda' & \mathbf{0} \\ \mathbf{0} & A_1 \end{bmatrix}, \quad and \quad V^T XX^T V = \begin{bmatrix} \Lambda & \mathbf{0} \\ \mathbf{0} & A_2 \end{bmatrix}$$

*for diagonal $r \times r$ matrices $\Lambda, \Lambda'$ and arbitrary $A_1 \in \mathbb{R}^{(k_0-r) \times (k_d-r)}, A_2 \in \mathbb{R}^{(k_0-r) \times (k_0-r)}$.*

*Proof.* Gradient descent on the objective

$$\arg\min_{f \in \mathcal{F}} \frac{1}{2n} \sum_{i=1}^n \|y^{(i)} - f(x^{(i)})\|_2^2$$

proceeds via the following update rule:

$$W_i^{(t+1)} = W_i^{(t)} + \frac{\gamma}{n}(W_d^{(t)} \cdots W_{i+1}^{(t)})^T \sum_{l=1}^n (y^{(l)} - f(x^{(l)}))(W_{i-1}^{(t)} \cdots W_1^{(t)} x^{(l)})^T, \quad \forall i \in [d],$$

$$(10)$$

where $\gamma$ is the learning rate and superscript $(t)$ denotes the gradient descent step. Assume that the network is initialized to be aligned, that is, there exist orthonormal $U_i, V_i$ and diagonal matrices $\Sigma_i$ such that $W_i = U_i \Sigma_i V_i^T$ and $U_i = V_{i+1}$ for $i \in [d-1]$. Substituting into Equation (10) yields

$$W_i^{(t+1)} = U_i \Sigma_i^{(t)} V_i^T + \frac{\gamma}{n} (U_d \Sigma_d^{(t)} \cdots \Sigma_{i+1}^{(t)} V_{i+1}^T)^T \sum_{l=1}^{n} (y^{(l)} - f(x^{(l)}))(U_{i-1} \Sigma_{i-1}^{(t)} \cdots \Sigma_1^{(t)} V_1^T x^{(l)})^T$$

$$= U_i \left( \Sigma_i^{(t)} + \frac{\gamma}{n} \prod_{j=i+1}^{d} \Sigma_j^{(t)T} U_d^T \sum_{l=1}^{n} (y^{(l)} - f(x^{(l)})) x^{(l)T} V_1 \prod_{j=1}^{i-1} \Sigma_j^{(t)T} \right) V_i^T$$

$$= U_i \left( \Sigma_i^{(t)} + \frac{\gamma}{n} \prod_{j=i+1}^{d} \Sigma_j^{(t)T} (U_d^T Y X^T V_1 - \Sigma_d^{(t)} \cdots \Sigma_1^{(t)} V_1^T X X^T V_1) \prod_{j=1}^{i-1} \Sigma_j^{(t)T} \right) V_i^T.$$

Thus strong alignment is an invariant if and only if for all $i$, the quantity

$$\prod_{j=i+1}^{d} \Sigma_j^{(t)T} (U_d^T Y X^T V_1 - \Sigma_d^{(t)} \cdots \Sigma_1^{(t)} V_1^T X X^T V_1) \prod_{j=1}^{i-1} \Sigma_j^{(t)T}$$

is an $k_i \times k_{i-1}$ diagonal matrix for all $t$. At initialization each of the $\Sigma_j$ have rank at least $r$. Considering $i = 1$ and $i = d$, the above quantity is diagonal if and only if the matrix

$$U_d^T Y X^T V_1 - \Sigma_d^{(t)} \cdots \Sigma_1^{(t)} V_1^T X X^T V_1 \tag{11}$$

has its top $r$ rows and top $r$ columns all diagonal; i.e. we can write this expression as

$$\begin{bmatrix} D & \mathbf{0} \\ \mathbf{0} & A \end{bmatrix} \tag{12}$$

for an $r \times r$ diagonal matrix $D$ and an arbitrary $(k_d - r) \times (k_0 - r)$ matrix $A$.

For the first direction, assume that strong alignment is an invariant, i.e. that Equation (11) can be written in the above block diagonal form. Define $\Sigma_{tot}^{(t)} = \Sigma_d^{(t)} \cdots \Sigma_1^{(t)}$ – this is a diagonal matrix whose only nonzero entries are the first $r$ on the diagonal. We know that

$$U_d^T Y X^T V_1 - \Sigma_{tot}^{(t)} V_1^T X X^T V_1$$

is of the form of Equation (12) for all gradient descent steps $t$, and thus the quantity

$$\left( \Sigma_{tot}^{(t)} - \Sigma_{tot}^{(0)} \right) V_1^T X X^T V_1$$

is of this form as well. Assuming that we've not initialized any of the singular values to be their optimal value (which is satisfied with probability 1), the top $r$ diagonal entries of $\Sigma_{tot}^{(t)} - \Sigma_{tot}^{(0)}$ are nonzero, which means that the top left $r \times r$ submatrix of $V_1^T X X^T V_1$ is diagonal, and that the top right submatrix consists of all zeros. But since $V_1^T X X^T V_1$ is symmetric, the bottom left submatrix must also consist of all zeros, and thus we have

$$V_1^T X X^T V_1 = \begin{bmatrix} D_2 & \mathbf{0} \\ \mathbf{0} & A_2 \end{bmatrix}$$

for an $r \times r$ diagonal matrix $D_2$ and arbitrary $(k_0 - r) \times (k_0 - r)$ matrix $A_2$. Plugging this into Equation (11) implies that $U_d^T Y X^T V_1$ must be of this form as well.

We next show the other direction. Assume that for some orthonormal matrices $U$ and $V$, it holds that $V^T X X^T V$ is diagonal and $U^T Y X^T V$ can be written in the block matrix form given by Equation (12). Initializing the layers such that $U_d = U, V_1 = V$, and $U_i = V_{i+1}$ for $i \in [d-1]$ implies that Equation (11) is also of this block diagonal form, as desired. □

## B  PROOF OF COROLLARY 1

*Proof.* The conditions of strong alignment imply the conditions of Lemma 2, which in turn implies that there exist orthonormal matrices $U, V$ such that

$$U^T Y X^T V = \begin{bmatrix} \Lambda' & \mathbf{0} \\ \mathbf{0} & A_1 \end{bmatrix}, \text{ and}$$

$$V^T X X^T V = \begin{bmatrix} \Lambda & \mathbf{0} \\ \mathbf{0} & A_2 \end{bmatrix},$$

where $\Lambda, \Lambda'$ are $r \times r$ diagonal matrices. Furthermore, from the proof of Theorem 1, if the layers are initialized to be aligned, with $U_d = U$ and $V_1 = V$, then the gradient descent updates are as follows:

$$W_i^{(t+1)} = U_i \left( \Sigma_i^{(t)} + \frac{\gamma}{n} \prod_{j=i+1}^{d} \Sigma_j^{(t)^T} (U^T Y X^T V_1 - \Sigma_d^{(t)} \cdots \Sigma_1^{(t)} V^T X X^T V) \prod_{j=1}^{i-1} \Sigma_j^{(t)^T} \right) V_i^T.$$

Since the minimum of the ranks of the $\Sigma_i^{(t)}$ is $r$, only the top $r$ singular values of $W_i$ are updated. Plugging in the expressions for $U^T Y X^T V$ and $V^T X X^T V$ and restricting to the top $r$ singular values (which we denote by $\Sigma_i'$), we obtain the statement of Corollary 1, with the singular values of each layer being updated as:

$$\Sigma_i'^{(t+1)} = \Sigma_i'^{(t)} + \frac{\gamma}{n} \prod_{j \neq i} \Sigma_j'^{(t)} (\Lambda' - \prod_{j=1}^{d} \Sigma_j'^{(t)} \Lambda).$$

This completes the proof. $\qquad\square$

## C  PROOF OF PROPOSITION 1

*Proof.* By Corollary 1, under strong alignment, each singular value is updated independently of each other. Thus we can focus on how the $k$th singular value for each layer is updated. Recall that $\sigma_k(W_i^{(t)})$ denotes the $k$th diagonal entry of $\Sigma_i^{(t)}$. Since we're focusing on a fixed $k$, we drop the subscript $k$ for convenience and let $\sigma_i^{(t)}$ equal $\sigma_k(W_i^{(t)})$. The $\sigma$ are updated by the following update rule:

$$\sigma_i^{(t+1)} = \sigma_i^{(t)} + \frac{\gamma}{n} \prod_{j \neq i} \sigma_j^{(t)} (\lambda_k' - \lambda_k \prod_{j=1}^{d} \sigma_j^{(t)}),$$

where $\lambda_k', \lambda_k$ are the $k$th diagonal elements of $\Lambda', \Lambda$. We assume that $\Lambda'$ and $\Lambda$ have the same zero pattern. Therefore $\lambda_k = 0$ if and only if $\lambda_k' = 0$. If both of these values are zero, then $\sigma_i$ is not updated.

Otherwise, assume $\lambda_k, \lambda_k' \neq 0$. Note that $\lambda_k > 0$, since $X X^T$ is positive semidefinite. We can also negate columns of $U$ to ensure that $\lambda_k' > 0$ as well. Let $\eta = \frac{\gamma \lambda_k}{n}$, and define $S^{(t)} = \prod_{j=1}^{d} \sigma_j^{(t)}$. This yields

$$\sigma_i^{(t+1)} = \sigma_i^{(t)} + \eta \frac{S^{(t)}}{\sigma_i^{(t)}} (\frac{\lambda_k'}{\lambda_k} - S^{(t)}). \tag{13}$$

Therefore (dropping the superscript to let $S = S^{(t)}$),

$$S^{(t+1)} = \prod_{i=1}^{d} \sigma_i^{(t+1)} = \prod_{i=1}^{d} \left( \sigma_i^{(t)} + \eta S \frac{1}{\sigma_i^{(t)}} (\frac{\lambda_k'}{\lambda_k} - S) \right)$$

$$= S + \sum_{T \subset [d]:|T| \geq 1} \eta^{|T|} S^{|T|} (\frac{\lambda_k'}{\lambda_k} - S)^{|T|} \prod_{i \in T} \frac{1}{\sigma_i^{(t)}} \prod_{i \notin T} \sigma_i^{(t)}$$

$$= S + \sum_{T \subset [d]:|T| \geq 1} \eta^{|T|} S^{|T|+1} (\frac{\lambda_k'}{\lambda_k} - S)^{|T|} \prod_{i \in T} \frac{1}{(\sigma_i^{(t)})^2},$$

and hence

$$\frac{\lambda_k'}{\lambda_k} - S^{(t+1)} = \frac{\lambda_k'}{\lambda_k} - S - \sum_{T \subset [d]:|T| \geq 1} \eta^{|T|} S^{|T|+1} (\frac{\lambda_k'}{\lambda_k} - S)^{|T|} \prod_{i \in T} \frac{1}{(\sigma_i^{(t)})^2} \tag{14}$$

$$= (\frac{\lambda_k'}{\lambda_k} - S) \left( 1 - \sum_{T \subset [d]:|T| \geq 1} \eta^{|T|} S^{|T|+1} (\frac{\lambda_k'}{\lambda_k} - S)^{|T|-1} \prod_{i \in T} \frac{1}{(\sigma_i^{(t)})^2} \right). \tag{15}$$

Thus we obtain

$$\frac{\lambda'_k}{\lambda_k} - S^{(t+1)} = \left(\frac{\lambda'_k}{\lambda_k} - S^{(t)}\right) \cdot r_k^{(t)}, \tag{16}$$

where

$$r_k^{(t)} = 1 - \sum_{T \subset [d]: |T| \geq 1} \eta^{|T|} S^{|T|+1} (\frac{\lambda'_k}{\lambda_k} - S)^{|T|-1} \prod_{i \in T} \frac{1}{(\sigma_i^{(t)})^2}. \tag{17}$$

We aim to bound $r_k^{(t)}$ from both above and below. First, we show that $r_k^{(t)}$ is nonnegative in order to prove the following lemma:

**Lemma 3.** $0 < S^{(j)} \leq \frac{\lambda'_k}{\lambda_k}$ for all $j \geq 0$.

*Proof.* We proceed by induction. By the original assumptions in Proposition 1, $0 < S^{(0)} \leq \frac{\lambda'_k}{\lambda_k}$. Now assume that $0 < S^{(j)} \leq \frac{\lambda'_k}{\lambda_k}$ for all $j \leq t$. By the update rule in Equation (13), $\sigma_i^{(j+1)} \geq \sigma_i^{(j)}$. Since $\sigma_i^{(0)} > 0$, $\sigma_i^{(j)} > 0$, so $S^{(j)} > 0$. We also have that

$$\prod_{i \in T} \frac{1}{(\sigma_i^{(t)})^2} \leq \prod_{i \in T} \frac{1}{(\sigma_i^{(0)})^2} \leq \frac{1}{(\min_i \sigma_i^{(0)})^{2|T|}}.$$

Next, note that we can bound

$$S^{|T|+1} (\frac{\lambda'_k}{\lambda_k} - S)^{|T|-1} \leq (\frac{\lambda'_k}{\lambda_k})^{2|T|}.$$

This means that we can upper bound the sum in Equation (17) as

$$\sum_{T \subset [d]: |T| \geq 1} \eta^{|T|} S^{|T|+1} (\frac{\lambda'_k}{\lambda_k} - S)^{|T|-1} \prod_{i \in T} \frac{1}{(\sigma_i^{(t)})^2} \leq \sum_{T \subset [d]: |T| \geq 1} \eta^{|T|} (\min_i \sigma_i^{(0)})^{-2|T|} (\frac{\lambda'_k}{\lambda_k})^{2|T|}$$

$$= \left(1 + \eta \cdot (\min_i \sigma_i^{(0)})^{-2} (\frac{\lambda'_k}{\lambda_k})^2\right)^d - 1.$$

Since $\gamma \leq \frac{n \ln 2}{d} \cdot \frac{\min_i (\sigma_i^{(0)})^2 \lambda_k}{\lambda'^2_k}$, we have that $\eta \leq \ln 2 \cdot \frac{\min_i (\sigma_i^{(0)})^2}{d} \cdot \frac{\lambda_k^2}{\lambda'^2_k}$, and thus the right-hand side of the above expression can be upper bounded by

$$\left(1 + \eta \cdot (\min_i \sigma_i^{(0)})^{-2}\right)^d - 1 \leq e^{d\eta(\min_i \sigma_i^{(0)})^{-2}} - 1 \leq e^{\ln 2} - 1 = 1.$$

Therefore $r_k^{(t)} \geq 0$. Plugging into Equation (16), since $S^{(t)} = S \leq \frac{\lambda'_k}{\lambda_k}$, we get that $S^{(t+1)} \leq \frac{\lambda'_k}{\lambda_k}$, which completes the inductive step. $\square$

Next, we would like to upper bound $r_k^{(t)}$ by a term independent of $t$ in order to obtain linear convergence. We can lower bound the sum in Equation (17) by the sets with size 1, so

$$\sum_{T \subset [d]: |T| \geq 1} \eta^{|T|} S^{|T|+1} (\frac{\lambda'_k}{\lambda_k} - S)^{|T|-1} \prod_{i \in T} \frac{1}{(\sigma_i^{(t)})^2} \geq \sum_{i=1}^d \eta S^2 \frac{1}{(\sigma_i^{(t)})^2} \geq \eta S^2 \cdot dS^{-2/d},$$

where the last inequality is due to AM-GM . Lemma 3 implies that $S^{(j+1)} \geq S^{(j)}$, which means that the above sum is at least $\eta d (S^{(0)})^{2-2/d}$, which means that we can upper bound $r_k^{(t)}$ by

$$r_k^{(t)} \leq 1 - \eta d (S^{(0)})^{2-2/d}.$$

This implies that $S^{(t+1)}$ is closer to $\frac{\lambda'_k}{\lambda_k}$ than $S$ is, and in particular

$$\frac{\lambda'_k}{\lambda_k} - S^{(t+1)} \leq (\frac{\lambda'_k}{\lambda_k} - S)(1 - d\eta(S^{(0)})^{2-2/d});$$

hence

$$\frac{\lambda'_k}{\lambda_k} - S^{(t)} \leq (\frac{\lambda'_k}{\lambda_k} - S^{(0)})(1 - d\eta(S^{(0)})^{2-2/d})^t.$$

Since the initialization is fixed, the quantity $1 - d\eta(S^{(0)})^{2-2/d}$ is fixed, and thus $S^{(t)}$ converges linearly to $\frac{\lambda'_k}{\lambda_k}$. Therefore each of the top $k$ singular values converge linearly to their optimal value $\frac{\lambda'_k}{\lambda_k}$, which means that the loss converges linearly as well.

To complete the proof, it suffices to show that this limit solution achieves a training loss of zero. This is proven in a more general setting at the end of Appendix E. $\qquad\square$

## D    Proof of Theorem 4

We can finally state the generalization of Theorem 1 to the non-square setting:

**Theorem 4.** *Let* $f : \mathbb{R}^{k_0} \to \mathbb{R}^{k_d}$ *be a linear fully connected network as in Equation equation 1, and let* $r = \min(k_0, \ldots, k_n)$*. Strong alignment is an invariant of training under the squared loss on the dataset* $(X, Y)$ *if and only if there exist orthonormal matrices* $U \in \mathbb{R}^{k_d \times k_d}, V \in \mathbb{R}^{k_0 \times k_0}$ *such that*

$$U^T Y X^T V = \begin{bmatrix} \Lambda' & \mathbf{0} \\ \mathbf{0} & A_1 \end{bmatrix}, \quad \text{and} \quad V^T X X^T V = \begin{bmatrix} \Lambda & \mathbf{0} \\ \mathbf{0} & A_2 \end{bmatrix}$$

*for diagonal* $r \times r$ *matrices* $\Lambda, \Lambda'$ *and arbitrary* $A_1 \in \mathbb{R}^{(k_0-r) \times (k_d-r)}, A_2 \in \mathbb{R}^{(k_0-r) \times (k_0-r)}$.

*Proof.* By Lemma 2 we know that under strong alignment there exist $U$ and $V$ satisfying the above conditions. In the other direction, Lemma 2 also tells us that given $U$ and $V$ satisfying the data conditions, all the conditions of strong alignment hold except for convergence to a global minimum.

To conclude, we must show that regardless of the zero pattern of $\Lambda$ or $\Lambda'$, under a strongly aligned initialization the network converges to a solution with a loss of zero.

Using the convenient notation that $\sigma_i^{(t)} = \sigma_k(W_i^{(t)})$, we again focus on how the $k$th singular values of each layer are updated, for some $k \in [r]$. Recall that the $\sigma$'s are updated as

$$\sigma_i^{(t+1)} = \sigma_i^{(t)} + \frac{\gamma}{n} \prod_{j \neq i} \sigma_j^{(t)} (\lambda'_k - \lambda_k \prod_{j=1}^d \sigma_j^{(t)}).$$

The rank of $X$ must be at least the rank of $Y$ in order for the data to be linearly interpolated. Therefore we can choosen $U, V$ (via permuting columns) to ensure that whenever $\lambda_k = 0, \lambda'_k = 0$ as well. This ensures that $\sigma_k(W_i^{(t)})$ is never updated. If $\lambda_k, \lambda'_k \neq 0$, then we showed in Proposition 1 that $S^{(t)}$ converges to $\lambda'_k/\lambda_k$ in the limit.

Finally, we consider the case where $\lambda'_k = 0, \lambda_k \neq 0$. Assume that $\sigma_i^{(t)} < 1$ and $\gamma < \frac{n}{\lambda_k}$. Then, the $\sigma_i$'s update as

$$\sigma_i^{(t+1)} = \sigma_i^{(t)} + \frac{\gamma}{n} \prod_{j \neq i} \sigma_j^{(t)} \left( -\lambda_k \prod_{j=1}^d \sigma_j^{(t)} \right) = \sigma_i \left( 1 - \eta \prod_{j \neq i} (\sigma_j^{(t)})^2 \right),$$

where $\eta = \frac{\gamma \lambda_k}{n}$. We observe that $0 \leq \sigma_i^{(t+1)} \leq \sigma_i^{(t)}$. Therefore

$$0 \leq S^{(t+1)} = S^{(t)} \prod_{i=1}^d \left( 1 - \eta \prod_{j \neq i} (\sigma_j^{(t)})^2 \right) \leq S^{(t)} \exp \left( -\eta \sum_{i=1}^d \prod_{j \neq i} (\sigma_j^{(t)})^2 \right) \leq S^{(t)} \exp \left( -\eta d S^{(t)2-2/d} \right).$$

Since $S^{(0)}$ is positive, we see that $0 \leq S^{(t+1)} \leq S^{(t)}$, and therefore $S^{(t)}$ must converge to some constant $c$. Assume that $c \neq 0$. For all $\epsilon > 0$, there exists some $t$ such that $S^{(T)} < c + \epsilon$. Then,

$$S^{(T+1)} \leq S^{(T)} \exp \left( -\eta d S^{(T)2-2/d} \right) < (c + \epsilon) \exp \left( -\eta c^{2-2/d} \right),$$

where $\exp\left(-\eta c^{2-2/d}\right)$ is a constant which is less than 1. Hence if we choose $\epsilon$ such that $\exp\left(-\eta c^{2-2/d}\right) < \frac{c+\epsilon}{c}$, then $S^{(T+1)} < c$, a contradiction. Therefore $c = 0$, and hence $S^{(t)} \to 0 = \lambda'_k/\lambda_k$.

In general, we have shown that if $\lambda_k \neq 0$, then $\sigma_k(W_1(t)) \cdots \sigma_k(W_d(t)) \to \lambda'_k/\lambda_k$. This solution is given by $f(x) = U_d\Lambda'\Lambda^{-1}V_1^T x$, which is the solution given by the pseudoinverse which obviously has a loss of zero. □

## E COMPLETING THE PROOF OF THEOREM 1

*Proof.* In Lemma 1, we showed that in the setting where all layers are square, alignment is equivalent to strong alignment. Theorem 4 states that in general, strong alignment is an invariant if and only if there exist $U, V$ satisfying particular data conditions. Since in the square setting $r = k$, by Theorem 4 we have that strong alignment is an invariant if and only if there exist $U, V$ such that $U^T Y X^T V$ and $V^T X X^T V$ are diagonal, as desired. □

## F ALIGNMENT FOR 1-DIMENSIONAL OUTPUTS

**Proposition 5.** *Assuming gradient descent avoids the point where all parameters are zero, alignment is an invariant of training for any linear fully connected network $f : \mathbb{R}^{k_0} \to \mathbb{R}$, any convex, twice continuously differentiable loss function, and data $(X, Y) \in \mathbb{R}^{k_0 \times n} \times \mathbb{R}^{1 \times n}$ for which the network can achieve zero training error.*

*Proof.* If we initialize the weight matrices to be rank 1 and aligned, then the matrices $\{\Sigma_i^{(t)}\}_{i=1}^d$ are diagonal with a single non-zero entry. Following the proof of Theorem 1, we obtain that alignment is an invariant if the matrix

$$\prod_{j=i+1}^{d} \Sigma_j^{(t)T} \left( U_d^T \sum_{k=1}^{n} \frac{\partial \ell}{\partial f}\bigg|_{(x^{(k)}, y^{(k)})} x^{(k)T} V_1^{(t)} \right) \prod_{j=1}^{i-1} \Sigma_j^{(t)T}$$

is diagonal. When $i \neq 1, d$, this matrix is clearly of rank 1 and diagonal (and has a single nonzero entry). This implies that $U_i, V_i$ are invariant for all $i \neq 1, d$. If $i = d$, then since $k_d = 1$, the above quantity is also a rank 1 diagonal matrix, implying that $U_d$ and $V_d$ are invariant. Finally, if $i = 1$, the above matrix is rank-1 but not necessarily diagonal. However, all but the top row are zeros, which after plugging into the gradient descent update rule implies that $U_1$ is invariant as well. Importantly, layers $W_{i+1}, W_i$ for $i \in [d-1]$ remain aligned regardless of the loss function used, as the expression above is always a diagonal matrix with a single nonzero entry when the layers are initialized to be rank 1. The final step is to show that training leads to zero error according to Definition 3. To do this, we first characterize the stationary points and then under assumptions, we prove that the loss converges to zero.

We now characterize the stationary points of the above update. Let $v_1^{(t)}$ denote the first column of $V_1^{(t)}$, and let $\sigma_1(W_j^{(t)})$ denote the top singular value in the usSVD of $W_j^{(t)}$. Then the stationary points are given by:

1. $\sigma_1(W_j^{(t)}) = 0$ for $j \in [d]$.

2. $v_1^{(t)} \perp \sum_{k=1}^{n} \frac{\partial \ell}{\partial f}\bigg|_{(x^{(k)}, y^{(k)})} x^{(k)T}$

If we initialize $\sigma_1(W_1^{(0)}) = 0$, then we have that:

$$\sigma_1(W_1^{(t)}) v_1^{(t)T} = \sum_{k=1}^{n} c_k^{(t)} x^{(k)T}$$

$$c_k^{(t+1)} = \sum_{k=1}^{n} \left( c_k^{(t)} + \gamma \prod_{j \neq k} \sigma_1(W_j^{(t)}) \frac{\partial \ell}{\partial f}\bigg|_{(x^{(k)}, y^{(k)})} \right) x^{(k)T}$$

for $c_k^{(t)} \in \mathbb{R}$ and $\forall t \in \mathbb{Z}_{\geq 0}$. Hence, updates to $v_1^{(t)}$ are in the span of the data, and so assuming that $\{x^{(k)}\}_{k=1}^n$ are linearly independent, $v_1^{(t)}$ cannot be orthogonal to $\sum_{k=1}^n \frac{\partial \ell}{\partial f}\Big|_{(x^{(k)}, y^{(k)})} x^{(k)^T}$ unless the $c_k^{(t)}$ are all 0, i.e. $\sigma_1(W_1^{(t)}) = 0$ for $t > 0$.

Next, if we initialize $\sigma_1(W_i^{(0)}) = \sigma_1(W_j^{(0)})$, then $\sigma_1(W_i^{(t)}) = \sigma_1(W_j^{(t)})$ for all $i, j \in \{2, \ldots d\}, t \geq 0$ since for all $i \in \{2, \ldots d\}$:

$$\sigma_1(W_i^{(t+1)}) = \sigma_1(W_i^{(t)}) + \prod_{j \neq i} \sigma_1(W_j^{(t)}) \left( \sum_{k=1}^n \frac{\partial \ell}{\partial f}\Big|_{(x^{(k)}, y^{(k)})} x^{(k)^T} v_1^{(t)} \right)$$

This initialization corresponds to layers $W_{i+1}, W_i$ being balanced for $i \in \{2, \ldots d\}$. Thus, under this initialization, the only other stationary point is given by $\sigma_1(W_i^{(t)}) = 0$ for all $i \in \{2, \ldots d\}$.

Hence, if gradient descent avoids the non-strict saddle points given by $\sigma_1(W_i^{(t)}) = 0$ for all $i \in \{2, \ldots, d\}$ and $\sigma_1(W_i^{(t)}) = 0$ for all $i \in [d]$, then gradient descent converges to a local (and thus global) minimum of the convex loss. The former stationary point can be avoided by re-parameterizing the network such that $\sigma_1(W_i^{(t)}) = \sigma_1$ for all $i \in \{2, \ldots d\}$ (i.e. $\sigma_1 = 0$ now corresponds to a strict saddle as defined in Lee et al. (2016)), and then taking a random initialization for $\sigma_1$. This would correspond to gradient descent on the original parameterization with a scaling factor on the learning rate for parameters $\sigma_1(W_i^{(t)})$ for $i \in \{2, \ldots d\}$. The latter stationary point is avoided by the assumption in the proposition. $\qquad \square$

## G    PROOF OF PROPOSITION 2

*Proof.* For any matrices $A, B \in \mathbb{C}^{m \times n}$, we have that $2\sigma_i(AB^*) \leq \sigma_i(A^*A + B^*B)$ (Bhatia, 1997). Thus letting $A = W_2, B = W_1^T$, we see that

$$2\sigma_i(W_2 W_1) \leq \sigma_i(W_2^T W_2 + W_1 W_1^T)$$
$$\implies 2\sum_i \sigma_i(P) \leq \sum_i \sigma_i(W_2^T W_2 + W_1 W_1^T)$$
$$= \|W_2^T W_2 + W_1 W_1^T\|_1$$
$$\leq \|W_2^T W_2\|_1 + \|W_1 W_1^T\|_1$$
$$= \|W_2\|_F^2 + \|W_1\|_F^2$$

This lower bound is in fact achieved for an aligned solution. If the SVD of $P$ is $P = U\Sigma V^T$, setting $W_1 = W\Sigma^{\frac{1}{2}} U^T$ and $W_2 = U\Sigma^{\frac{1}{2}} V^T$ yields $\|W_1\|_F^2 = \|W_2\|_F^2 = \text{Tr}(\Sigma)$, so $\|W_1\|_F^2 + \|W_2\|_F^2 = 2\text{Tr}(\Sigma)$. $\qquad \square$

## H    PROOF OF THEOREM 2

*Proof.* Given an arbitrary loss function, assume that the $i$th layer is restricted to some structure given by a subspace $\mathcal{S}$ and basis matrices $A_1, \ldots A_m$, so that at timestep $t$ we have that

$$W_i^{(t)} = \sum_{j=1}^m (c_j^i)^{(t)} A_j$$

We take the gradient of the loss with respect to the $c_j^i$. The chain rule yields:

$$\frac{\partial l}{\partial c_j^i} = \sum_{p,q=1}^n \frac{\partial l}{\partial (W_i)_{pq}} \cdot \frac{\partial (W_i)_{pq}}{\partial c_j^i} = \sum_{p,q=1}^n \frac{\partial l}{\partial (W_i)_{pq}} \cdot A_{pq}^j$$

The gradient descent update on $c_j^i$ is thus:

$$(c_j^i)^{(t+1)} = (c_j^i)^{(t)} - \eta \cdot \frac{\partial l}{\partial c_j^i} = (c_j^i)^{(t)} - \eta \sum_{p,q=1}^n \frac{\partial l}{\partial (W_i)_{pq}} \cdot A_{pq}^j$$

The corresponding update on $W^i$ becomes

$$\begin{aligned}
W_i^{(t+1)} &= \sum_{j=1}^m (c_j^i)^{(t+1)} A_j \\
&= \sum_{j=1}^m (c_j^i)^{(t)} A_j - \eta \sum_{j=1}^m \sum_{p,q=1}^n \frac{\partial l}{\partial (W_i)_{pq}} \cdot A_{pq}^j A^j \\
&= W_i^{(t)} - \eta \sum_{j=1}^m \sum_{p,q=1}^n \frac{\partial l}{\partial (W_i)_{pq}} \cdot A_{pq}^j A^j
\end{aligned}$$

We calculate the projection operator $\pi$ of some arbitrary matrix $M$ onto $\mathcal{S}$. We can write

$$\pi(M) = \sum_{j=1}^m \frac{\langle M, A^j \rangle A^j}{\|A_j\|_2^2} = \sum_{j=1}^m \sum_{p,q=1}^m \frac{M_{pq} A_{pq}^j A^j}{\|A_j\|_2^2}.$$

If we define the operator $\pi_{\mathcal{S}}$ as

$$\pi_{\mathcal{S}}(M) = \sum_{j=1}^m \langle M, A^j \rangle A^j = \sum_{j=1}^m \sum_{p,q=1}^m M_{pq} A_{pq}^j A^j,$$

then gradient descent on the $c$ gives the following update rule on the $W^i$:

$$W_i^{(t+1)} = W_i^{(t)} - \eta \cdot \pi_{\mathcal{S}} \left( \frac{\partial l}{\partial W_i} \right).$$

If the $A_j$ all have norm 1, then, $\pi = \pi_{\mathcal{S}}$, and this is the same update rule given by projected gradient descent with respect to the subspace $\mathcal{S}$. Otherwise, $\pi_{\mathcal{S}}$ is simply the projection $\pi$ followed by appropriate scaling in each of the basis directions. $\qquad\square$

## I   TREATING A CONVOLUTIONAL LAYER AS A LINEAR SUBSPACE

Consider a $3 \times 3$ image. We map it to a 9-dimensional vector as follows

$$\begin{bmatrix} x_1 & x_2 & x_3 \\ x_4 & x_5 & x_6 \\ x_7 & x_8 & x_9 \end{bmatrix} \implies \begin{bmatrix} x_1 & x_2 & x_3 & x_4 & x_5 & x_6 & x_7 & x_8 & x_9 \end{bmatrix}^T.$$

Then, the linear transformation given by applying the $3 \times 3$ convolutional filter $\begin{bmatrix} c_1 & c_2 & c_3 \\ c_4 & c_5 & c_6 \\ c_7 & c_8 & c_9 \end{bmatrix}$ is given by the matrix

$$W = \begin{bmatrix}
c_5 & c_4 & 0 & c_2 & c_1 & 0 & 0 & 0 & 0 \\
c_6 & c_5 & c_4 & c_3 & c_2 & c_1 & 0 & 0 & 0 \\
0 & c_6 & c_5 & 0 & c_3 & c_2 & 0 & 0 & 0 \\
c_8 & c_7 & 0 & c_5 & c_4 & 0 & c_2 & c_1 & 0 \\
c_9 & c_8 & c_7 & c_6 & c_5 & c_4 & c_3 & c_2 & c_1 \\
0 & c_9 & c_8 & 0 & c_6 & c_5 & 0 & c_3 & c_2 \\
0 & 0 & 0 & c_8 & c_7 & 0 & c_5 & c_4 & 0 \\
0 & 0 & 0 & c_9 & c_8 & c_7 & c_6 & c_5 & c_4 \\
0 & 0 & 0 & 0 & c_9 & c_8 & 0 & c_6 & c_5
\end{bmatrix}.$$

Then $\mathcal{S}$ consists of all matrices of the form $W$. $\mathcal{S}$ is a 9-dimensional subspace of $\mathbb{R}^{9 \times 9}$, with an orthonormal basis with coefficients being the $c_i$.

## J    PROOF OF PROPOSITION 3

*Proof.* For $i \in [d]$, let $U_i \Sigma_i V_i^T$ be a usSVD of $W_i$ witnessing alignment of $f$. We can then rewrite $Y = f(X)$ as $Y = U_d \prod_{i=1}^{d} \Sigma_i V_1^T X$, thus proving the desired statement. □

## K    PROOF OF PROPOSITION 4

Before we can prove Proposition 4, we require the following definition from combinatorics.

**Definition 5.** *A partition of an integer $k$ is a tuple $\lambda = (\lambda_1, \ldots, \lambda_s)$ such that $\lambda_i \geq \lambda_{i+1}$ for all $i$ and $k = \lambda_1 + \cdots + \lambda_s$. Each $\lambda_i$ is called a* part *of $\lambda$. We let $s(\lambda)$ denote the number of parts of $\lambda$ and we write $\lambda \vdash k$ to indicate that $\lambda$ is a partition of $k$.*

*Proof of Proposition 4.* Given a $k \times k$ matrix $A$, let $\lambda(A)$ denote the partition $\lambda$ of $k$ such that $\lambda_i$ is the multiplicity of the $i^{\text{th}}$ greatest singular value of $A$. Let $U(A)$ denote the set of matrices $U$ such that $U \Sigma V^T$ is a usSVD of $A$. The dimension of $U(A)$ is

$$\sum_{i=1}^{s(\lambda(A))} \binom{\lambda_i}{2}.$$

To see this, note that any orthonormal basis of the eigenspace of $AA^T$ corresponding to the multiplicity-$\lambda_i$ eigenvalue of $AA^T$ can be the corresponding columns in an element of $U(A)$ and that the set of orthonormal bases of an $m$-dimensional linear space is $\binom{m}{2}$.

For any set $Q$ of matrices, Define $U(Q)$ to be the set of all possible sets of left-singular vectors of elements of $S$. That is,

$$U(Q) := \bigcup_{A \in Q} U(A).$$

For each partition $\lambda$ of $k$, let $T_\lambda$ denote the set of matrices $A$ such that $\lambda(A) = \lambda$. The dimension of $T_\lambda \cap S$ is at most $r$ and therefore the dimension of $U(S \cap T_\lambda)$ is at most

$$r + \sum_{i=1}^{s(\lambda)} \binom{\lambda_i}{2}.$$

Let $\mathcal{O}(k, n)$ denote the set of $k \times n$ matrices with orthonormal columns. Assume alignment is possible over $S$ for a non-measure-zero set of matrices with $n$ columns. Then there exists $B \subseteq \mathcal{O}(k, n)$ with $\dim(B) = \dim(\mathcal{O}(k, n))$ such that for every $U' \in B$, $U(S)$ contains a matrix whose first $n$ columns are $U'$. Therefore $\dim(U(S)) \geq \dim(\mathcal{O}(k, n))$. Since $\dim(\mathcal{O}(k, n)) = \binom{k}{2} - \binom{k-n}{2}$, the following must be satisfied for some $\lambda \vdash k$

$$r + \sum_{i=1}^{s(\lambda)} \binom{\lambda_i}{2} \geq \binom{k}{2} - \binom{k-n}{2}. \tag{18}$$

This is attained when $\lambda = (k)$, but in this case $T_\lambda$ is simply the set of scalar multiples of the identity. If we forbid $\lambda = (k)$, then we claim that the maximum value of $r + \sum_{i=1}^{s(\lambda)} \binom{\lambda_i}{2}$ is attained by $\lambda = (k-1, 1)$. To see this, note that for all $p < q$,

$$\binom{q-p}{2} + \binom{p}{2} = \binom{q}{2} - p(q-p) < \binom{q}{2}.$$

For $p > 0$, this is maximized when $p = 1$. This implies that the maximum value of $\sum_{i=1}^{s(\lambda)} \binom{\lambda_i}{2}$ will be obtained in as few summands as possible (which in our case is two), and in particular when $\lambda_1 = k - 1$ and $\lambda_2 = 1$. In this case, equation 18 becomes

$$r + \binom{k-1}{2} \geq \binom{k}{2} - \binom{k-n}{2}.$$

Taking the logical negation of the above inequality and simplifying gives $r < k - 1 - \binom{k-n}{2}$. □

## L  ADDITIONAL EXPERIMENTS

We provide the following empirical evidence demonstrating that when the conditions of Theorem 1 are satisfied, invariance of alignment can indeed be observed empirically. We use a 2-hidden layer fully connected network with 9 hidden units per layer.

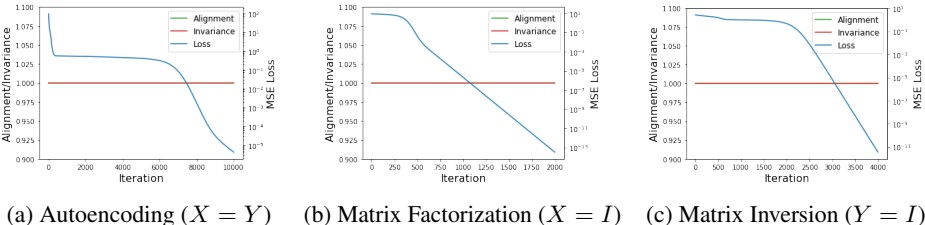

(a) Autoencoding $(X = Y)$    (b) Matrix Factorization $(X = I)$    (c) Matrix Inversion $(Y = I)$

Figure 3: As proven in our work, alignment is an invariant of training when $X, Y$ satisfy the conditions of Theorem 1.

## M  EXPERIMENTAL SETUP

We provide network architectures and hyperparameters used for our experiments below. We trained our networks on an NVIDIA TITAN RTX GPU using the PyTorch library. In all settings, we train using gradient descent with a learning rate of $10^{-2}$ until the loss was below $10^{-4}$.

1. Figure 1a: We use a 2-hidden layer fully connected network with 9 hidden units per layer. Our data is given by matrices $(X, Y) \in \mathbb{R}^{9 \times 9}$ where each matrix entry is drawn from a standard normal distribution.

2. Figure 1b: We use a 2-hidden layer fully connected network with 1024 hidden units in the first hidden layer and 64 hidden units in the second hidden layer. Our data consists of 256 linearly separable examples from MNIST and is trained using Squared Loss.

3. Figure 1c: We use a 2-hidden layer fully connected network with 1024 hidden units in the first hidden layer and 64 hidden units in the second hidden layer. Our data consists of 256 linearly separable examples from MNIST and is trained using Cross Entropy Loss.

4. Figure 2a: We use a 2-hidden layer network with 4 hidden units per layer, where each layer is constrained to be a Toeplitz matrix. Our input $X$ is equal to the identity, and our output $Y$ is a $4 \times 4$ matrix with each entry sampled from a standard normal distribution.

5. Figure 2b: We use a 2-hidden layer convolutional network with a single $3 \times 3$ filter in each layer, stride of 1, and padding of 1. Our data consists of a single example from MNIST.

Code for the experiments can be found at the following anonymized github link: `https://anonymous.4open.science/r/33277cc0-6074-46c4-8642-7feadd678278/`.

