# OpenReview forum: "On Alignment in Deep Linear Neural Networks"
_ICLR.cc/2021/Conference — Reject_

### Official Review · AnonReviewer1 · 2020-10-25
**A characterization of alignment in deep linear networks**

**Rating:** 4
**Confidence:** 4

**Review:**

This paper considers deep linear networks trained by gradient descent with the squared loss, and characterizes when alignment happens, meaning that for any pair of adjacent weight matrices W_{i+1} and W_i, the (unsorted, signed) right singular vectors of W_{i+1} are identical to the (unsorted, signed) left singular vectors of W_i. This paper further gives a few examples where the conditions of alignment hold, and proves a convergence rate for aligned networks. For networks with constrained layer structure, such as convolutional networks, this paper shows that aligned networks in general cannot achieve zero training error with the squared loss. Finally, empirical support of the theory is provided.

I think this paper provides a nice necessary and sufficient condition for alignment in deep linear networks, which simplifies the training dynamic and let us prove a linear convergence rate. On the other hand, when the layer structure is constrained, it is shown that aligned networks in general cannot achieve zero training error, which is interesting.

However, there are some weaknesses that should be noticed:
1. As given by Definition 3, if alignment is an invariant of training, it only means that there exists a good initialization. To get a practical algorithm, we need to find such a good initialization, which seems nontrivial. For example, matrix factorization and inversion are mentioned as an example which satisfies the conditions of Theorem 1, but according to the proof of Theorem 1, to factorize or inverse a matrix M, we need to initialize the first and last weight matrices using an unsorted, signed singular value decomposition of M, which does not make sense.
2. Theorem 1 requires the input dimension to be the same as the output dimension, and it only uses square layers, which I think means that each hidden width is also equal to the input dimension. This setting is pretty restrictive; for example, autoencoding is mentioned as an example of Theorem 1, but if the hidden widths are equal to the input dimension, then there seems to be no point to do such an encoding. On the other hand, Corollary 1 doesn't seem to need the hidden widths to be identical, which is inconsistent.

Additionally, the appendices should be provided.

---

> ### Author Response · Authors · 2020-11-19
> **Response to Reviewer 1**
>
> We thank the reviewer for their feedback, and address their concerns below:
>
> We would first like to clarify that the appendices were indeed provided in the initial submission, and can be viewed if you click the zip download next to the supplementary material (which was presented as an option in the Author FAQ (https://iclr.cc/Conferences/2021/AuthorGuide).
>
> * “To get a practical algorithm, we need to find such a good initialization, which seems nontrivial”
>     * We would like to emphasize that the purpose of studying gradient descent in linear neural networks is not to devise a practical algorithm for training such networks, but rather to provide intuition on non-convex optimization more generally.
>     * As mentioned in our related work, our definition of invariance of alignment is the same as used in various prior works, e.g. Gidel et. al. (2019), Saxe et. al. (2018), Saxe et. al. (2014), and also the following submission to ICLR 2021 (https://openreview.net/forum?id=D9pSaTGUemb), where this is referred to as “Spectral Initialization”. In particular, our invariance of alignment definition corresponds to the assumption used in Theorem 3 from Gidel et. al. (2019) to study discrete training dynamics and prove linear convergence of gradient descent in a two layer linear network. In our paper, we first establish necessary and sufficient conditions on training data under which this assumption holds (Theorem 1) and then extend the convergence result from Gidel et. al. (2019) to networks of arbitrary depth (Proposition 2).
>     * Since the purpose of analyzing the training dynamics of linear neural networks is to gain intuition around how gradient descent can converge to a global minimum in non-convex settings, our work shows that when initialized to be aligned, deep linear neural networks converge linearly to a global minimum. We feel that our analysis of these assumptions from prior work provide an important positive result relevant to the study of linear neural networks. In addition, we believe it is equally important to understand the limitations of assumptions used in prior works and we therefore argue that our negative results are also of high relevance.
> * “Theorem 1 requires the input dimension to be the same as the output dimension”
>     * Before the statement of Theorem 1 we clearly state “To simplify notation, we consider the case when the layers are square matrices, i.e. $k_i = k_j$ for all $0 \le i, j \le d$. The general result for non-square matrices is provided in Appendix D.”

---

> > ### Comment · AnonReviewer1 · 2020-11-22
> > **The notion of alignment is too strong**
> >
> > Thanks for the response. I am sorry that I did not notice the supplementary file; indeed a general version of Theorem 1 is provided in Appendix D.
> >
> > However, I still think the notion of alignment is too strong; therefore even though Theorem 1 gives a sufficient and necessary characterization of alignment, its importance is still unclear to me. As I mentioned above, for the matrix factorization and inversion examples, to initialize the network so that Theorem 1 can be applied, we already need to know an unsorted, signed singular value decomposition of the matrix, which is enough to solve the problem. I do not see why such an initialization is reasonable. For example, for linear margin maximization with linear networks, only a very mild condition on the initialization is needed, and specifically the solution is not encoded in the initialization.

---

### Official Review · AnonReviewer4 · 2020-10-29
**Alignment notion too stringent, limited application.**

**Rating:** 4
**Confidence:** 4

**Review:**

This article extends the notion of alignment [Ji and Telgarsky, 2018] to linear neural networks with multiple output nodes, which requires the consecutive layers (i+1, i) to have the same (right, left) singular spaces. The authors identify necessary and sufficient conditions under which, alignment is an invariant of the gradient descent iterates (Definition 3), which in particular means that the gradient descent iterates only update the singular values of the layers, and not their singular vectors. The authors studied alignment for several shallow and deep linear architectures, and specify learning rates for which gradient descent enjoys exponential convergence.

Here are my main comments, mostly about the significance of the results.
Alignment (Definition 2), itself, is a very restrictive assumption: it requires the left and right singular spaces of all consecutive layers to be aligned, i.e. V_{i+1} = U_{i}. In fact, the authors show in Theorem 3 that alignment cannot occur, for a large class of interesting architectures, including convolutional neural networks. Therefore, studying this notion is not well-motivated to begin with.
Alignment being an invariant of training (Definition 3), is a far more stringent assumption, particularly because it requires the singular spaces of all hidden layers to remain fixed during the training. As it is shown in Theorem 1, this property holds if and only if the input and the output have the same right singular space. On the other hand, when this condition can be satisfied, e.g. for instances that are given in section 4.2, it is not clear if the analysis provides any additional insights/improvements over the previous works.

Some additional comments:
Definition 3 requires interpolation under the linear model, i.e., the data is clean (no noise), and the relationship between the output and the input is completely characterized by a linear map, which makes the result less interesting from a practical view. While this setting is well-studied in the literature, this work does not provide comparisons against the previous works.
Matrix Sensing in section 4.2: how can matrix sensing be an instance of a deep linear network with a multi-dimensional output? Both the labels y_i and the network predictions Tr(M_i^T W_d...W_1) are scalers, and hence 1-dimensional.

To sum up, the paper studies an extension of alignment [Ji and Telgarsky, 2018] to linear networks with multi-dimensional output. This notion is too stringent -- as the authors confirm in the paper -- and cannot be satisfied unless in some special cases. On the other hand, when the condition can be satisfied, it is not clear if the results provide any insights/improvements over the previous work. For these reasons, I vote for rejecting this submission.


========================
Final Recommendation

I have read the rebuttal and decided to keep my score. I think this study needs to be further motivated.
I also want to clarify a minor issue in authors rebuttal. In contributions, you say: "We demonstrate that alignment is an invariant for fully connected networks with multidimensional outputs only in special problem classes including autoencoding, matrix factorization and matrix sensing. This is in contrast to networks with 1-dimensional outputs, where there exists an initialization...". My point is that the matrix sensing problem that you study here has 1-dimensional output.

---

> ### Author Response · Authors · 2020-11-19
> **Response to Reviewer 4**
>
> We thank the reviewer for their feedback, and address their concerns below:
> * “Therefore, studying this notion is not well-motivated to begin with.”
>     * As mentioned in our related work, our definition of invariance of alignment is the same as used in various prior works, e.g. Gidel et. al. (2019), Saxe et. al. (2018), Saxe et. al. (2014), and also the following submission to ICLR 2021 (https://openreview.net/forum?id=D9pSaTGUemb), where this is referred to as “Spectral Initialization”. In particular, our invariance of alignment definition corresponds to the assumption used in Theorem 3 from Gidel et. al. (2019) to study discrete training dynamics and prove linear convergence of gradient descent in a two layer linear network. In our paper, we first establish necessary and sufficient conditions on training data under which this assumption holds (Theorem 1) and then extend the convergence result from Gidel et. al. (2019) to networks of arbitrary depth (Proposition 2).
>     * Since the purpose of analyzing the training dynamics of linear neural networks is to gain intuition around how gradient descent can converge to a global minimum in non-convex settings, our work shows that when initialized to be aligned, deep linear neural networks converge linearly to a global minimum. We feel that our analysis of these assumptions from prior work provide an important positive result relevant to the study of linear neural networks. In addition, we believe it is equally important to understand the limitations of assumptions used in prior works and we therefore argue that our negative results are also of high relevance.
> * “...the relationship between the output and the input is completely characterized by a linear map, which makes the result less interesting from a practical view. While this setting is well-studied in the literature, this work does not provide comparisons against the previous works”
>     * All of the works we discuss extensively in our related work section (Ji & Telgarsky (2018), Gidel et al. (2019), Saxe et al. (2014, 2019), Arora et al. (2018), Du et al. (2018)) rely on the assumption that the output is a linear function of the input.
>     * In fact, virtually all previous work on linear neural networks assumes that a linear network can perfectly interpolate the data. This is because we are interested in the dynamics of gradient descent in the overparameterized regime, where there are infinitely many solutions which perfectly fit the data and we would like to understand which of these solutions is learned by the network.
> * “ Matrix Sensing in section 4.2: how can matrix sensing be an instance of a deep linear network with a multi-dimensional output?”
>     * As written, the matrix sensing case involves parameters that are all multi-dimensional $W_1, W_2, \ldots W_d$.

---

### Official Review · AnonReviewer2 · 2020-10-30
**The paper studies the concept of alignment in deep neural networks in the context of linear networks with multidimensional outputs.**

**Rating:** 7
**Confidence:** 3

**Review:**

The paper presents an extension of the idea of alignment in linear neural networks, that can help in providing convergence analysis of such networks. Such a notion was previously studied for networks with a single output. The current paper extends it to networks with multi-dimensional outputs. The paper offers multiple interesting results: a) conditions on the datasets where alignment can remain invariant b) lack of alignment or invariance for networks with constrained layers

The paper is very clearly written and is offers a coherent explanation of the ideas presented. The presented theoretical results are interesting. However, the constraints on the datasets X and Y in Theorem 1 are pretty stringent. It might be interesting to present a study on how realistic these conditions are in practice.

---

> ### Author Response · Authors · 2020-11-19
> **Response to Reviewer 2**
>
> We thank the reviewer for their feedback, and their positive comments. We address their concerns:
> * “However, the constraints on the datasets X and Y in Theorem 1 are pretty stringent. It might be interesting to present a study on how realistic these conditions are in practice.”
>     * In Section 4.2, we discuss common problems for which the data condition is satisfied, including autoencoding, matrix factorization/inversion, matrix sensing, and networks with 1d outputs.
>     * Furthermore, we note that these constraints on the data are implicit in various prior works e.g. Gidel et. al. (2019), Saxe et. al. (2018), Saxe et. al. (2014).

---

### Official Review · AnonReviewer3 · 2020-10-31
**Insufficient motivation for the studying the strong definition of "alignment invariance"**

**Rating:** 4
**Confidence:** 5

**Review:**

The paper introduces the property of *alignment invariance* of gradient descent for linear networks. Under this definition (Definition 3) the left singular vectors of the linear transform in layer i is aligned with right singular vectors of i+1, and further the definition requires that the singular vectors remain constant throughout the training process (only the singular values are updated along gradient updates). The main theorem derives the necessary and sufficient conditions under which such a notion of alignment invariance is possible. The paper further shows that when such a condition holds, the trajectories of gradient descent are simplified for analysis of convergence speed.

My main concern with the paper is that the definition of alignment invariance in Def 3 is very restrictive and the motivation for considering such strong conditions is not sufficiently justified.
1. This definition is significantly different and a much stronger condition compared to similar  properties in prior work: (a) “alignment” in Ji & Telgarsky 2018 (for 1D output) refers to condition where only the final converged solution has aligned singular vectors, but at initialization and throughout training the alignment and invariance need not (and does not) hold, and (b) the balancedness in Arora et al. 2018 and Du et al. 2018 (for gradient flow) are conditions on W_i^TW_i^T-W_{i+1}W_{i+1} being small throughout training. This requires that the singular values be nearly aligned but they need not be constant through the training process. This is a significant difference as for any dataset, the balancedness W_i^TW_i^T-W_{i+1}W_{i+1} can be shown to always be an invariant through gradient flow but that is not true for singular vectors.

2. In practice, this condition almost always never holds without requiring circular computations. For example, in order to initialize an autoencoder to satisfy Def 3, one needs to do know the spectral decomposition of data matrix X, which is in fact the primary computation performed by linear autoencoders! Also, No examples are provided where predictors obtained through such restrictive flows lead to useful models.

3. It is ok to consider restrictive assumptions if such assumptions provide significant new insights and results that are atleast somewhat relevant in practice. However the main use case for this Def 3 in the paper is that these conditions make the analysis of convergence speed of gradient descent simpler. But for linear networks, even though the underlying problem is non-convex, many existing results show convergence of gradient descent under often much less restrictive conditions, e.g., with just balancedness at initialization in Arora et al. 2019a. More specifically, the paper shows Def 3 leads to simplified dynamics, but it is not demonstrated how one can use the simplified dynamics to derive new results/insights that were not previously known. Finally, in terms of pure analysis, although the paper extends for deeper networks, the key proof ideas are similar to those of Gidel et al. 2019.

---

> ### Author Response · Authors · 2020-11-19
> **Response to Reviewer 3**
>
> We thank the reviewer for their feedback, and address their concerns below:
>
> * “This definition is significantly different and a much stronger condition compared to similar properties in prior work”
>     * As mentioned in our related work, our definition of invariance of alignment is the same as used in various prior works, e.g. Gidel et. al. (2019), Saxe et. al. (2018), Saxe et. al. (2014), and also the following submission to ICLR 2021 (https://openreview.net/forum?id=D9pSaTGUemb), where this is referred to as “Spectral Initialization”. In particular, our invariance of alignment definition corresponds to the assumption used in Theorem 3 from Gidel et. al. (2019) to study discrete training dynamics and prove linear convergence of gradient descent in a two layer linear network. In our paper, we first establish necessary and sufficient conditions on training data under which this assumption holds (Theorem 1) and then extend the convergence result from Gidel et. al. (2019) to networks of arbitrary depth (Proposition 2).
>     * Since the purpose of analyzing the training dynamics of linear neural networks is to gain intuition around how gradient descent can converge to a global minimum in non-convex settings, our work shows that when initialized to be aligned, deep linear neural networks converge linearly to a global minimum. We feel that our analysis of these assumptions from prior work provide an important positive result relevant to the study of linear neural networks. In addition, we believe it is equally important to understand the limitations of assumptions used in prior works and we therefore argue that our negative results are also of high relevance.
> * “In practice, this condition almost always never holds without requiring circular computations.”
>     * We would like to emphasize that the purpose of studying gradient descent in linear neural networks is not to devise a practical algorithm for training such networks, but rather to provide intuition on non-convex optimization more generally.
> * “many existing results show convergence of gradient descent under often much less restrictive conditions, e.g., with just balancedness at initialization”
>     * We would like to emphasize that balancedness is a restrictive condition, as it assumes that the singular values between layers are equal. Our definitions do not assume anything about the singular values of each layer.

---

### Decision · Program_Chairs · 2021-01-07
**Final Decision**

**Decision:**

Reject

**Comment:**

The consensus view was that the reviewers were not convinced that the analysis done in the paper was sufficient motivated.